# Contexts for developing of national essential diagnostics list. Lessons from a mixed-methods study of existing documents, stakeholders and decision making on tier-specific essential in-vitro diagnostics in African countries

**Winny Koster** [1]*, **Elishebah Maruta Mutegi** [2], **Francis Ocen** [2], **Collins Otieno Odhiambo** [2], **Maina Michael Waweru** [2], **Albert Gautier Ndione** [3], **Sri Lakshmi Priyanka Yerra** [4], **Jenny L. Grunwald** [5], **Delores Mack** [5], **Kekeletso Kao** [6], **Lucy A. Perrone** [7,8,9], **Pascale Ondoa** [1,2]

**1** Amsterdam Institute for Global Health and Development, University of Amsterdam, Amsterdam, the Netherlands, **2** African Society for Laboratory Medicine, Addis Ababa, Ethiopia, **3** Department of Sociology, Faculty of Arts and Humanities, Cheikh Anta Diop University, Dakar, Senegal, **4** Department of Epidemiology, School of Public Health, University of Washington, Seattle, Washington State, United States of America, **5** School of Public Health, Health Systems and Population Health, University of Washington, Seattle, Washington State, United States of America, **6** FIND, Geneva, Switzerland, **7** Department of Global Health, University of Washington, Seattle, Washington State, United States of America, **8** Department of Laboratory Medicine, University of Washington, Seattle, Washington State, United States of America, **9** Department of Pathology and Laboratory Medicine, Faculty of Medicine, University of British Columbia, Vancouver, Canada

\* winnykoster@gmail.com

## Abstract

Since 2019, the WHO recommends the development and implementation of National Essential Diagnostics List (NEDL) to facilitate availability of In-Vitro Diagnostics (IVDs) across the various tiers of the healthcare pyramid, facilities with or without a laboratory on-site. To be effective, the development of NEDL should take into consideration the challenges and opportunities associated with current modalities for organization of tier specific testing services in-country. We conducted a mixed-methods analysis set out to explore available national policies, guidelines and decision-making processes that affect accessibility of diagnostics in African countries; 307 documents from 48 African countries were reviewed and 28 in-depth (group) interviews with 43 key-informants in seven countries were conducted between June and July 2022. Of the 48 countries, Nigeria was the only one with formal NEDL. Twenty-five countries had national test menus (63% outdated, from 2015 or earlier) all specifying tests by laboratory tier (5 including the "community tier"), with additional details on equipment (20), consumables (12), and personnel requirements (11). The most popular criteria to select essential IVDs in the quantitative analysis relate to specificities of the tests, whereas in the qualitative study most mentioned were health care and laboratory contextual factors. Quality assurance and waste management for tests at "community tier" were highlighted as concerns by all the respondents. Additional barriers to

**Data Availability Statement:** The data underlying our findings have been uploaded as supplementary files in S1, S2, S3, S4, S5 and S6.

**Funding:** The study was funded by FIND from an award of the Australian Government; by Vital strategy through the RESOLVE to Save Lives grant and by the Bill and Melinda Gates Foundation: BMGF OPP1162196 and INV-003603. WK, EMM, FO, AGN, KK and LAP received salaries or funding from the FIND Australian Government award. COO, FO, MMW and PO received salaries from BMGF. The funders had no role in the study design, data collection and analysis, decision to publish, or preparation of the manuscript.

**Competing interests:** The authors have declared that no competing interests exist.

implementation included the low decision-making power of Laboratory Directorates within the Ministry of Health, as well as the chronic budgetary gaps for clinical laboratory services and policy and strategic plan development outside of vertical programmes. Four countries out of seven would rather revise their test menus by updating them and add "community tier", than developing a separate NEDL, the former being considered more operational. This study provides a unique set of practical recommendations to the process of development and effective implementation on NEDL in Africa.

## Introduction

Three major global health priorities, Universal Health Coverage (UHC), Antimicrobial Resistance, and Global Health Security, require diagnostics for implementation and population level surveillance. However, studies by the Lancet Commission on diagnostics show that 47% of the global population has little to no access to diagnostics and that diagnostics are underfunded because their central importance is underappreciated. The studies also conclude that 1.1 million deaths annually could be averted by better diagnostic access for six key conditions and that each dollar invested in diagnostics returns multiple dollars in benefits [1].

The literature reports low availability of diagnostics in Low and Middle Income Countries (LMIC), particularly in primary care settings: diagnostic availability normally increases in higher level health care facilities [2–4]. Even for priority public health diseases, the availability and quality of tests is low [3]. A study, analysing national laboratory policies and plans in sub-Saharan African countries, found that these problems are associated with several factors, including lack of budget available for the provision of laboratory services or for the implementation of laboratory improvement strategies [4]. Lack of a practical framework for monitoring and evaluating laboratory services and the implementation of national laboratory policies in most countries means that there are no indicators to measure the performance of laboratory services, including the availability of diagnostics [4].

Literature indicates that presently the laboratory systems are weak in many African countries and the status of laboratory services low. Within national MoHs, laboratory services are often not prioritised, and their management may be divided among several departments, resulting in a lack of budgeting, planning and diagnostics accountability [5]. Laboratory services suffer from lack of sufficiently qualified human resources at various levels, poor supply chains for diagnostics and erratic supply and maintenance of laboratory equipment [2, 6, 7].

International agencies support the laboratory domain, but their support, including supply of IVDs, is not evenly spread across laboratories and health problems. They have been prioritising specific disease programmes such as HIV/AIDS, tuberculosis, and malaria. This has made diagnostics for these diseases more readily available, including at the primary health care level, and has led to significant improvements in diagnosis of these conditions in LMICs [2]. In addition, more recently the coronavirus disease 2019 (COVID-19) pandemic has caused international organisations and donors to realise the urgency of availability and access to high-quality diagnostics [8].

With signing the 2008 Maputo Declaration on Strengthening of Laboratory Systems, African countries and partners recognized the need to develop quality-assured laboratory services as part of a greater framework of health system strengthening within resource-limited settings. Limited laboratory capacity was acknowledged as a major barrier to implementation and sustainability of prevention, treatment and care programs for HIV, malaria and tuberculosis [9]. To advance availability and access to diagnostics in African countries, on 16 November 2017,

the Africa Centres for Disease Control and Prevention launched the Africa Collaborative Initiative to Advance Diagnostics to promote the diagnostics agenda in the African region through better coordinated and synergized efforts that align with the priorities of MoHs. This partnership includes, among others, African Society for Laboratory Medicine (ASLM), the Institute for Health Research, Epidemiological Surveillance and Training, WHO Regional Office for Africa, the Clinton Health Access Initiative (CHAI), the African Field Epidemiology Network and UNITAID [10].

The WHO Essential Diagnostics List (EDL), launched in 2018, aims to make essential In-Vitro Diagnostics (IVD), including biochemical, pathology and microbiology tests, more widely available and accessible, which would contribute to the UHC programme, outbreak preparedness and health promotion. "Essential diagnostics" are those that satisfy the priority health care needs of the population and are selected with due regard to disease prevalence, public health relevance, evidence of efficacy and accuracy and comparative cost–effectiveness. The WHO EDL lists IVDs for two tiers, i) health facilities with and ii) facilities without laboratory on-site (latter includes community outreach health care). WHO urges countries to develop a national essential diagnostics list (NEDL). A pamphlet and technical document give guidelines how to adapt the WHO EDL to a country's local priority diseases, health care needs and resources and to possibly add granularity to the tiers, reflecting the country's health-care pyramid [11, 12].

When developing their NEDL, countries could base the selection of essential IVDs on the national disease burden, the national priority diseases list, or the National Essential Medicines List (NEML). However, in many African countries, data on national disease burden are not available or are unreliable [13, 14]. Because national priority diseases lists are difficult to establish, countries could draw on various global lists to establish their national lists. There is also the International Health Regulations surveillance list [15], which has 10 basic priority tests to diagnose diseases, but not all tests are relevant to all countries and others, such as antimicrobial resistance tests, are missing. It is therefore important that the specific realities of each country be considered to set up national lists of priority diseases that will allow the rational development of an NEDL. Furthermore, for successful NEDL implementation, all elements of the laboratory system must be in place and functional and thus considered when creating the list [14].

Another influential national document for selecting IVDs for an NEDL would be the NEML. Most countries have an updated NEML with the medicines needed to address the national priority health problems. An NEDL based on the NEML would ensure availability of medicines to prevent, manage or cure diagnosed diseases and conditions [6]. Ideally, disease management and diagnostic services should be aligned by tier of health care facility. Lack of (sufficiently) qualified human resources at various levels, poor supply chains for diagnostics and erratic supply and maintenance of laboratory equipment mean that this alignment is not always easy to attain [2, 6].

WHO advises that the first step for NEDL development is to constitute an MoH senior level committee comprised of key stakeholders from across the healthcare sector, and for this committee to appoint a technical committee that will be responsible for reviewing all applicable national documentation on IVDs and preparing a list of candidate IVD tests categories (e.g. name of the test, assay format(s), assay purpose and specimen type) towards developing the NEDL. Suggested committee members would include: personnel from the MoH or another national or regional authority involved in the development of national or regional policies and diagnostics guidelines; specialists and technical experts in the various areas of IVDs; experts in clinical laboratory, anatomical pathology and blood bank operations; and experts in evidence synthesis and appraisal, evidence-based medicine and health technology assessment [11, 12].

Prior to the WHO EDL 1st edition (2019) [16], many countries have already rationalized the use of diagnostics at different tiers of the national health care and laboratory network through a variety of operational and normative guidance documents recommending minimal testing packages. That guidance can be found under various document names, including: 'Standardisation of diagnostics', 'Harmonisation policies for laboratories', 'National test menu' or 'National norms for laboratory services'.

Thus, in the development of an NEDL one could learn from the content of existing documents on IVDs and from experiences with development of these documents. To this end, ASLM the Foundation for Innovative New Diagnostics (FIND) commissioned a study with main objective: to collect information from African countries on existing national guidance documents and on decision making when selecting tier-specific IVDs that could be the basis for recommendations to supporting countries in developing an NEDL or a similar guidance document. The specific study objectives were:

- to map and analyse content of relevant existing documents on IVDs by health care/laboratory tier

- to explore stakeholders and decision-making processes in development of existing guidance documents on IVDs

- to give recommendations for NEDL development.

## Materials and methods

The study used a mixed-method design, triangulating quantitative and qualitative data collection and data. The "quantitative arm" implied a desk review of documents possibly addressing tier specific diagnostic testing from 55 African Member State countries. Data collection in the "qualitative arm" was through in-depth semi-structured interviews with key informants in seven countries.

### Study questions

Following are the study questions for this paper, with in brackets whether they were addressed in the quantitative arm (QUANT) or/and qualitative arm (QUAL).

1. What are the categories of documents defining tier-specific IVDs and what items are included in these documents? (QUANT and QUAL)

2. Who are the stakeholders involved in development of national test menus and NEDL? (QUANT and QUAL)

3. What are the bases/criteria to assign essential IVDs to various tiers of the health care/laboratory system? (QUANT and QUAL)

4. To what extent are the countries' tier-defined national test menus aligned to the IVDs recommended in WHO EDL 2 for the diagnostics of HIV, tuberculosis, malaria, syphilis, cholera and the determination of full blood count? (QUANT)

5. What is the in-country knowledge and appreciation of the WHO EDL and its application at national level? (QUAL)

6. What are recommendations to MoH and other stakeholders (national and international) for (guidance in) developing and implementing an NEDL? (QUAL)

## Sampling and sample size of countries, documents and key-informants

**Quantitative arm.** An online search was conducted for key documents most likely to contain information on IVDs. A content analysis template with key search terms was created in line with the study questions, using Google search engine to search for documents on country's MoH and laboratory services websites, government websites, and PubMed/journals. From the online search, initially 362 documents were found from 55 AU member states. After cascading, 292 documents from 44 countries remained for full content review. Because of time constraints, countries were dropped for which we found less than 3 documents. Ten of the documents were national test menus, one an NEDL. The study team approached ASLM contact persons in the countries for which they did not find national test menus online and asked whether they had such documents; 15 countries reported they had and provided these documents, thus a total of 25 test menus were found. Four of the 15 countries providing test menus were not part of the initial sampling cascade; adding them to the original 44 brings the total number of countries included in the quantitative study to 48. (S1 Fig) S1 Data includes the quantitative data set; S1–S3 Tables the documents' URLs as far as available. Table 1 gives a breakdown of the category of documents in the quantitative analysis.

In this paper we mainly present findings on the national test menus, because these are closest to an NEDL. We analyse whether laboratory-based diagnostics by tier is encompassed in the documents, including testing in the community, what other items are included, which type of stakeholders are mentioned to have been involved in development of the document, which criteria for test selection, are mentioned and how do the listed IVDs for selected tracer disease and conditions (HIV, malaria, tuberculosis, syphilis, cholera, and full blood count) align to those in WHO EDL 2 [17]. (We started analysis of alignment before WHO EDL 3 was published, in 2021 [18].

**Qualitative arm.** Sampling of countries and respondents for the qualitative study was purposeful and was informed by the online desk review provisional findings (before the 15 additional test menus were collected through the ASLM contact persons) and in consultations between the desk review team and the two consultants leading the qualitative arm (social science researchers). The qualitative study sampled seven from the 28 (out of total 44) countries that had at least one document addressing IVDs by tier level, or were in the process of developing such document because these countries could share experiences of how decisions were made to assign IVDs to tiers. Sampled countries were Burkina Faso, Cameroon, Ethiopia, Kenya, Nigeria, Uganda, and Zimbabwe.

**Table 1. Documents reviewed by category.**

| Category | Number | % (N = 307) |
|---|---|---|
| National health policy and/or strategic plan | 65 | 21.2% |
| National HIV strategy (or plan) | 43 | 14.0% |
| National malaria strategy (or plan) | 40 | 13.1% |
| Others | 36 | 11.7% |
| National tuberculosis strategy (or plan) | 31 | 10.1% |
| National tests menus/norms/standardisation of diagnostics | 25 | 8.1% |
| Non-communicable diseases strategy (or plan) | 25 | 8.1% |
| Maternal, new born and child health strategy (or plan) | 22 | 7.2% |
| National laboratory policy and/or strategic plan | 16 | 5.2% |
| National health laboratory procurement plan | 3 | 1.0% |
| National Essential Diagnostic List | 1 | 0.3% |
| Total | 307 | 100% |

Study participants were decision-makers and other stakeholders (potentially) involved in the selection of IVDs for national guidance documents. In each country, LabCop country representatives [19] informed about the study, populated a table with names and background of potential key-informants. The two consultants interviewed the respondents by using a semi-structured interview guide tailored to the country and key-informant. Themes in the guide followed the study questions and included: laboratory system and challenges; stakeholders in the laboratory system; national documents with guidelines on IVDs; criteria for prioritising IVDs; intention and plans to develop and implement an NEDL.

The consultants conducted 28 (group) interviews between 7th June and 25th July 2022, involving 43 participants, 19 women and 24 men. For five countries they conducted interviews by Zoom conferencing; in two countries face-to-face (Kenya and Uganda). Respondents were staff from MoH national laboratory services (21); international partners (9); hospital laboratories (6); consultants for the Nigerian NEDL (3); professional associations (2); MoH Clinical Services Department (1); staff from regulatory body (1).

Two research assistants transcribed the interviews' audio-recordings in Microsoft Word documents. The consultants analysed the data in three steps: i) entering the answers of multiple interviews and document analysis into one country specific question guide; ii) summarizing the country findings in a topic-matrix by country–analysing differences and similarities from different data sources; iii) comparing the similarities and differences by theme across countries and explaining differences from country contexts (See qualitative data base in S2 Data).

## Ethics

Ethical approval for the study was granted by Solution IRB, Yarnell, United States of America, under Protocol #2022/03/6. Potential respondents for the qualitative study received information on the study and the question guide by email and were requested to participate and, if they were willing, to indicate time-availability. Before start of the interview, they were asked their (oral) consent for audio-recording. Only two respondents did not agree to audio-recording and two respondents asked to pause recording when they mentioned potentially sensitive issues (related to critiquing others). Respondents received the final draft report of the qualitative study and were asked to verify the information for their country. Research assistants removed the audio and WORD files from their computers after they submitted the transcriptions approved by the consultants.

## Results

### Existing documents on tier-specific IVDs

Thirty five/48 countries (35/48) have at least one document that addresses tier-specific testing. One country (Nigeria) developed her National Essential Diagnostic List (NEDL) following WHO EDL structure, while 25 countries have a national document aimed at standardizing/harmonizing testing across the laboratory network (for short: "test menu"). The other nine have one or more documents with tier specific IVDs, often for vertical disease programmes. Eleven countries have documents that define testing, but tests are not specified by laboratory/health facility tier. For two countries (Congo and Comoros) we found documents that also focused on laboratory development, but these did not define tests (Table 2).

The test menus date from 2005–2021 (median 2013). Most documents for vertical disease programmes are more recent 1999–2025 (median 2019) than the test menus. The vertical disease programme policies and strategic plans are regularly revised–we used the most recent update for our analysis.

**Table 2. Countries' progress in defining IVDs by tier of the diagnostic network.**

| Has NEDL | Has a national test menu that standardises/harmonises testing by tier | Has document(s) that defines testing per the tiered network | Has document(s) that defines testing, but not by tier | Has documents on laboratories but not defining testing |
|---|---|---|---|---|
| Nigeria | Benin<br>Botswana<br>Burkina Faso<br>Burundi<br>Cameroon<br>Côte d'Ivoire<br>eSwatini<br>Ethiopia<br>Gabon<br>Gambia<br>Ghana<br>Guinea<br>Guinea Bissau<br>Kenya<br>Liberia<br>Malawi<br>Mali<br>Niger<br>Senegal<br>Sierra Leone<br>South Africa<br>Tanzania<br>Togo<br>Uganda<br>Zimbabwe | Angola<br>CAR<br>DRC<br>Lesotho<br>Mauritania<br>Mauritius<br>Namibia<br>Somalia<br>Zambia | Cabo Verde<br>Chad<br>Djibouti<br>Equatorial Guinea<br>Madagascar<br>Mozambique<br>Rwanda<br>São Tomé and Príncipe<br>Seychelles<br>South Sudan<br>Sudan | Congo<br>Comoros |

Focusing on the 25 national test menus (Table 3) we see that in contrast to the WHO EDL and the Nigerian NEDL, that specify tests for two tiers (with and without laboratory on-site), all national test menus specify the IVDs for different laboratory and/or health facility tiers. Only five of the 25 test menus (20%) address testing at the "community tier" (including testing in health facilities without laboratory on-site and in the community), as the WHO NEDL guidelines advise, and the Nigerian NEDL does.

Vertical disease programme documents with testing guidelines more often identify IVDs for the community tier: 55% of malaria documents addressing IVDs (N = 31), 32% of such HIV documents (N = 28) identified IVDs for community tier [7]. In all seven countries of the qualitative study, community health workers and/or staff in health posts and health centres without a laboratory do rapid diagnostic tests (RDTs), usually for malaria, HIV, pregnancy, and albumin and protein (with urine dipsticks); some of these tests are specified in vertical disease programme documents. The Ethiopian respondents reported that health extension workers and staff of health posts without laboratories perform malaria RDTs and pregnancy tests.

**Table 3. Summary of items included in national test menus (N = 25).**

| Item | In # included | In % included |
|---|---|---|
| Tests by health facility/laboratory tier | 25 | 100% |
| Equipment by laboratory tier | 20 | 80% |
| Consumables by laboratory tier | 12 | 48% |
| Personnel specified by laboratory tier | 11 | 44% |
| Equipment brands and/or selection considerations | 7 | 28% |
| Tests for "community tier" | 5 | 20% |
| Consumable brands and/or selection considerations | 5 | 20% |

These workers are trained, and their tests' quality monitored by laboratory professionals of health centres with a laboratory.

The 25 test menus are more extensive than a standard NEDL because in addition to defining testing by laboratory tier, 21 include guidelines for equipment, 12 for consumables, and 11 for qualifications of laboratory personnel that are required for the tests (Table 3). Test menus of five countries (Burundi, Gabon, Liberia, Sierra Leone and Zimbabwe) address all three items. Seven (out of 25) countries, specify recommended equipment brands and/or selection considerations. The Malawi and Zimbabwe test menus are the most elaborate in specifying the equipment brands and explaining the selection considerations, including technical specifications, supply chain considerations, infrastructure requirements, availability of service engineers, and costs associated with procuring and operating the machine. The Malawi test menu categorises three levels of operational considerations in: Critical (e.g. equipment assessed, in-country, by CDC, WHO etc. and report available), important and desirable (e.g. machine will run as single platform) and additional (e.g. sample transportation). The Zimbabwe test menu has two levels of requirements for equipment: i) critical requirements (e.g. amongst others equipment comes with warranty (1–2 years) and spares guarantee from the manufacturer, services engineers available) and ii) other requirements (e.g. can use alternate sources of power such as battery or solar and equipment is in current production, not end-of-life).

In five countries test menus include brand names of consumables and/or selection criteria. The Kenyan test menu mentions the most extensive selection criteria for specific brands. In Botswana the test menu classifies consumable products into either fast moving or slow moving.

## Stakeholders in development of test menus

Table 4 presents quantitative data on stakeholder involvement in the development of the test menus. Nineteen test menus mentioned the stakeholders involved in the development.

**Table 4. Stakeholders involved in development of test menus (multiple responses) (N = 19).**

| Stakeholder | Number of documents that mention the stakeholders' involvement | % of documents that mention the stakeholders (N = 19) |
|---|---|---|
| MoH Health care providers from facilities (laboratory staff, clinicians) | 15 | 79% |
| Implementing partners | 14 | 74% |
| Funding agency | 13 | 68% |
| MoH—Disease programs | 12 | 63% |
| MoH—Laboratory department (administrative) | 12 | 63% |
| Technical agencies | 12 | 63% |
| MoH—Department not specified | 11 | 58% |
| Training/research institutions | 11 | 58% |
| MoH—Reference laboratories | 10 | 53% |
| MoH—Pharmacy department | 6 | 32% |
| MoH—Regional and National Laboratories | 6 | 32% |
| Professional Health Associations | 6 | 32% |
| MoH—Procurement agency/national warehouse | 5 | 26% |
| MoH—Clinical services | 3 | 16% |
| MoH—Preventive services | 3 | 16% |
| Others* | 3 | 16% |
| MoH—Epidemiology | 2 | 11% |

* National Army; civil society; political leaders.

The top-six of stakeholder categories mentioned in over 63% of test menus were: MoH health care providers, MoH Disease programmes, MoH Laboratory Services, international implementing, funding and technical partners. The professional associations were mentioned to be involved in only 32% of test menus. In the Nigerian NEDL development, all stakeholder categories were involved. It should be noted that we cannot conclude on stakeholder involvement in test menus, because six test menus out of 25 (24%) did not list the stakeholders.

In all seven countries of the qualitative study, the same international stakeholders are involved for technical, financial and implementation support to the laboratory sector. The documents and respondents cited WHO, United States Agency for International Development (USAID), United States CDC, and ASLM as technical support partners. For implementation CHAI, ASLM, Africa CDC, United States President's Emergency Plan for AIDS Relief (PEP-FAR), USAID, and United States CDC are identified as the main partners. The usual stakeholders for funding are The Global Fund to Fight AIDS, Tuberculosis and Malaria, World Bank (WB), FIND, CHAI, USAID, United Kingdom Aid Direct, and United Nations Children's Emergency Funds (UNICEF). In addition to these cross-cutting international partners, other mentioned stakeholders are country-specific [7].

The qualitative study, exploring stakeholders and their decision-making power, found that laboratory technical working groups with a wide range of national and international stakeholder members exist in Burkina Faso, Uganda, and Nigeria. These working groups have been or will be involved in developing documents with IVD guidelines. Nigeria is the only country with an established and powerful National Laboratory Technical Working group, inaugurated by the Minister of Health on 27 January 2017, with membership of all organisations and associations in the laboratory domain (53 members) that has its secretariat at the Federal MoH Laboratory Services section. In the four other countries of the qualitative study, no formalised technical committees on laboratories exist. However, in Ethiopia, Kenya and Zimbabwe, ad-hoc teams or working groups consisting of staff from the Laboratory Services, some with technical assistance from partners, develop(ed) laboratory harmonisation guidelines, policies and other documents.

Qualitative data show that the MoH Laboratory Services section was always involved in the process of developing of documents related to IVDs. Its position in the MoH–being a department with a budget, or a sub-section without budget–would influence how much decision-making power the section has. Apart from Uganda, where the Department of National Health Laboratory and Diagnostic Services is an MoH department, in all other countries the Laboratory Services are sub-divisions of other MoH departments, or a laboratory wing of an autonomous public health institute (the Ethiopian Public Health Institute). Several respondents from the Laboratory Services mentioned they had no mandate and/or resources to implement or revise existing test menus. However, not only its position within the MoH, but also its connection with and support by the Minister of Health and international funding and technical partners was found to play a role in the Laboratory Services decision-making power. In Nigeria, where the Laboratory Services are a sub-department, a strong connection and keen interest of the Minister of Health and these international partners in developing the laboratory system has facilitated developing the NEDL. The partners paid a team of four national consultants to write the NEDL, in constant deliberation with the Laboratory Services and National Laboratory Technical Working Group.

Several respondents pointed at important stakeholders to involve in the discussions for more probable take up of the list: i) implementing partners of (smaller) donor agencies that are not passing through the MoH Laboratory Services, so to align them with official national laboratory policies and guidelines; ii) clinicians, because they request the tests; iii) regulatory and litigation agencies.

The Nigerian experience indicates that already in NEDL development it is key to involve all stakeholders in future NEDL implementation, including regulators, government procurement offices and state-level laboratory services. Although the Nigerian NEDL is not anchored yet in the national laboratory strategic plan, and still many hurdles have to be taken before wide implementation, respondents reported that the NEDL is already used by the groups of stakeholders involved in its development, including the Medical Laboratory Science Council of Nigeria's Public Health IVD Control laboratory which now prioritizes testing and validating of IVDs that are listed in the NEDL.

## Criteria for selecting essential IVDs

Quantitative analysis of the 25 test menus and the NEDL, show that nine (eight test menus and the NEDL) contained information on criteria for test selection. These criteria mainly relate to specificities of the tests and less to contextual factors (Table 5).

The most important selection criterion mentioned is the sensitivity, specificity and susceptibility of tests, relying on WHO recommendations. Four countries report to have considered the priority health needs and disease burden of the population across specific regions of the country. Nigeria studied this in their laboratory landscape survey, which was input for the NEDL. Some of the selection criteria are interrelated. For instance, the possible relative complexity/simplicity of test criterion is related to available supplies and personnel at a specific tier. And, in cognisance of funding problems, Kenya and Uganda made sub-categories of priority IVDs: Vital, Essential, and Necessary.

Respondents in the qualitative study mentioned as most important considerations for selecting priority IVDs the health care and laboratory contextual factors, including type and capacities of laboratory personnel present; type and capacities of clinical personnel present and level of care provided; alignment with the NEML; accessibility of health care facilities and laboratories (affecting the ease of regularly transporting inputs for IVDs). Many also mentioned disease burden and priority health care needs of the population; vertical disease programmes; and cost of tests. It appears that in selecting tests for the test menus the countries already considered feasibility of implementation. Ugandan and Kenyan respondents elaborated on the three priority categories of IVDs: Vital, Essential, and Necessary (VEN criteria), by saying this was done with an eye on procurement agents and funders, reasoning that funding always is a problem. A respondent in Uganda explained: "*If national stores get money, they will focus on Vital. If they have more money, they can go to Essential. If they have even more money, they must go to Necessary*". On the question whether gender was one of the criteria for selecting IVDs, most respondents answered something like: '*not really, because generally diseases are not gender specific*". However, all countries consider pregnancy related tests (gender specific) essential. The Kenya and Ethiopia teams specified that a few other gender specific tests were on their priority IVD list: for cervical and breast cancer. The Ethiopia team

**Table 5. Selection criteria mentioned in eight test menus and NEDL (multiple response).**

| Test Selection criterion | No. of documents with mention (N = 9) |
|---|---|
| Sensitivity/specificity/performance | 6 |
| Complexity/simplicity | 5 |
| Priority health needs, disease burden, potential emergency situation | 4 |
| Available human resource, IVDs and supplies by level | 3 |
| Affordability/Cost effectiveness | 3 |
| Reliability / Safety / Quality | 2 |

expressed that they would consider gender as a criterion for prioritisation in the NEDL they were currently developing.

## Alignment of IVDs in national test menus with WHO EDL 2

The WHO guidelines on NEDL development state that countries can decide which IVDs to select from the WHO EDL and which to drop or add and at which laboratory tier to use them, depending on their epidemiology, human resources and infrastructure [12]. Our study reviewed the 25 test menus and the Nigerian NEDL for their alignment to the WHO EDL 2 concerning IVDs for tracer diseases and conditions, e.g. HIV infection, tuberculosis, malaria, syphilis, cholera, and full blood count. We reviewed the test menus for the two tiers mentioned in the WHO EDL 2: The community tier (in 5 test menus and the NEDL) and the medical laboratory tier (in 25 test menus and the NEDL). It should be noted that some of the IVDs recommended by WHO could not have been on the older test menus because they were only approved after the date of the test menu.

The alignment between WHO EDL 2 recommendations for selected tests, and the actual national test menus or NEDL is shown in Table 6 (for community tier, N = 6 countries) and Table 7 (laboratory tier; N = 26 countries).

At the "community tier", Nigeria was the only country covering the full set of selected tests in its national EDL document, as well as providing information on the assay format. Five other countries considered diagnostic test for the community, including the serology tests for the diagnosis of HIV infection (eSwatini, Sierra Leone, Togo), qualitative nucleic acid testing for early infant diagnostic (eSwatini, Sierra Leone), malaria diagnostic (Sierra Leone) and Syphilis (Togo, Uganda). Beside the recommendation from the NEDL of Nigeria, 11 out of the 13 test recommendations did not provide details on the assay format. HIV self-testing, CD4 count, Crypotococcal antigen and cholera diagnostic tests were largely left out from all national recommendations, except for Nigeria. In the Kenyan test menu, none of the tracer tests is considered; the Kenyan test menu for community level only lists tests for glucose.

National strategies for testing at the laboratory tiers indicated that haematology assays (blood count) was included in test menus from all 25 countries and the Nigerian NEDL. The next most popular tests in national guidance are the HIV serology screening tests (24 and 25 of 26 countries), the CD4 count (24/26, suggesting that countries largely select CD4 enumeration to be done in a laboratory facility despite available POC format) and malaria microscopy (25/ 26). Conventional microscopy or culture-base tuberculosis diagnostics were included in 23 and 24 of 26 country guidelines, respectively. Modern molecular techniques for the detection and characterization of M tuberculosis remained rather confidential, included by 14 countries (NAAT for the detection of active TB and resistance) and by only 4 countries for LAMP. Tests for the management of advanced HIV disease were notably excluded by more than 50% of country guidance: cryptococcal antigen; histoplasma antigen, TB LAM and interferon Gamma ELISPOT.

The test menus of most countries partially or completely aligned with most of the WHO recommended IVDs that were approved prior to their development. This was mostly true for less complex tests like rapid antigen/antibody tests, microscopy and automated complete blood count but much less for complex molecular based-tests. For example 25/26 (96%) test menus were fully/partially aligned with the WHO EDL recommended anti-HIV rapid diagnostic test while only 14/26 (54%) test menus were aligned to the recommended Qualitative HIV virological nucleic acid test. The same observation is true for tuberculosis tests. Most test menus developed from 2018 (e.g. Ghana, Togo, Tanzania, South Africa) were more aligned to the WHO EDL recommended tests compared to those developed before development of the

**Table 6. Alignment between national test menus and WHO EDL 2 for IVDs recommended at "community tier".**

| Programs/ Diseases | Mentioned in WHO EDL 2 | | | Year of WHO approval | Mentioned in national test menus and NEDL | | | | | |
|---|---|---|---|---|---|---|---|---|---|---|
| | Diagnostic test | Test purpose | Assay format | | ESwatini | Kenya | Nigeria | Sierra Leone | Togo | Uganda |
| | | | | | 2010 | 2015 | 2021 | 2015 | 2021 | 2017 |
| **HIV** | HIV 1/2 antibody (anti-HIV Ab) | HIV self-testing | RDT | 2015 | Grey | Grey | Green | Grey | Grey | Grey |
| | HIV 1/2 antibody (anti-HIV Ab) | To diagnose HIV infection: adults, adolescents, children and infants > 18 months of age | RDT | 1985 | Rose | Grey | Green | Rose | Rose | Grey |
| | Combined HIV antibody/ p24 antigen (anti-HIV/p24 Ag) | For the diagnosis of HIV infection: adults, adolescents, children and infants > 18 months of age | RDT | | Rose | Grey | Green | Rose | Rose | Grey |
| | Qualitative HIV virological nucleic acid test | For diagnosis of HIV infection in infants < 18 months of age | Point-of-care nucleic acid test | | Rose | Grey | Green | Rose | Grey | Grey |
| | CD4 cell enumeration | Staging advanced HIV disease and for monitoring response to antiretroviral therapy | Point-of-care flow cytometry platform | 2009 | Grey | Grey | Green | Grey | Grey | Grey |
| | Cryptococcal antigen | Screening and diagnosis of cryptococcal meningitis in people with advanced HIV disease | RDT | 2009 | Grey | Grey | Green | Grey | Grey | Grey |
| **Tuberculosis (TB)** | Tuberculin skin (Mantoux) test (TST) | Diagnosis of latent TB infection | Intradermal test | 2015 | Grey | Grey | Green | Grey | Grey | Grey |
| **Malaria** | Plasmodium spp. antigens; species specific (e.g. HRP2) and/or pan-species specific (e.g. pan-pLDH) | Diagnosis of one or more human malaria species (P. falciparum, P. vivax, P. malariae, P. ovale) | RDT | 2006 | Grey | Grey | Green | Green | Grey | Grey |
| **Cholera** | Vibrio Cholerae antigen | Initial detection or exclusion of a cholera outbreak | RDT | 1993 | Grey | Grey | Green | Grey | Grey | Grey |
| **Syphilis** | Antibodies to Treponema Pallidum | Diagnosis of *T. pallidum* | RDT | 2013 | Grey | Grey | Green | Grey | Green | Rose |

Green: Test and test assay format mentioned in document

Rose: only diagnostic test or test purpose mentioned in document

Grey: No mention of the test in document

WHO EDL guidelines. The level of uptake of tests for laboratory tiers in national testing strategies positively correlated with earlier dates of recommendation by WHO (R = 0.700; p<0.001, spearman correlation data in S4 Table).

## Knowledge and appreciation of the WHO EDL and NEDL initiative

Respondents in the qualitative study got to know about the WHO EDL and the advice to develop an NEDL in international workshops, but not all had seen or read the WHO EDL. "*It is on my reading table*" said a respondent in Zimbabwe. Only in Ethiopia, a team from the Laboratory Services was already in the process of developing their NEDL. Four countries do not intend to develop a separate NEDL, but might use the WHO guidelines in revision of their test menus, which they consider more extensive. Zimbabwe has already put this in their 2022 workplan.

**Table 7. Alignment between national test menus and WHO EDL 2 for IVDs recommended at laboratories tiers.**

| Programs/ Diseases | Diagnostic test (Mentioned in WHO EDL 2) | Test purpose | Assay format | Year of WHO approval |
|---|---|---|---|---|
| HIV | Antibodies to HIV-1/2 (anti-HIV Ab) | For the diagnosis of HIV infection: adults, adolescents, children and infants > 18 months of age | RDT | 1999 |
| | Combined HIV antibody/p24 antigen (anti-HIV/p24 Ag) | For the diagnosis of HIV infection: adults, adolescents, children and infants > 18 months of age | RDT | |
| | | | Immunoassay | |
| | Qualitative HIV virological nucleic acid test | For diagnosis of HIV infection in infants < 18 months of age | Nucleic acid test | |
| | Quantitative HIV virological nucleic acid test | For monitoring response to antiviral treatment. For diagnosis of HIV infection in infants < 18 months of age (only if validated by the manufacturer) | Nucleic acid test (DBS) | |
| | CD4 cell enumeration | For staging advanced HIV disease For monitoring response to antiretroviral therapy. (In settings where viral load is not available) | Flow cytometry | 2009 |
| | Cryptococcal antigen | For screening and diagnosis of cryptococcal meningitis in people with advanced HIV disease | RDT | 2009 |
| | | | Immunoassay | 2009 |
| | Histoplasma antigen | To aid in the diagnosis of disseminated Histoplasmosis | Immunoassay | 2017 |
| TB | Mycobacterium tuberculosis bacteria | For diagnosis, treatment and monitoring of active TB | Microscopy | 2008 |
| | | For diagnosis and treatment monitoring of active TB including drug resistant TB | Bacterial culture | |
| | M. tuberculosis DNA | For diagnosis of active TB and simultaneous detection of rifampicin resistance | Nucleic acid test | 2008 |
| | | For diagnosis of active TB | Loop-mediated isothermal amplification (LAMP) | |
| | M. tuberculosis DNA mutations associated with resistance | For detection of resistance to first-line anti-TB medicines | Molecular line probe assay (LPA) | |
| | | For detection of resistance for second-line anti-TB medicines | Molecular line probe assay (LPA) | |
| | Drug susceptibility testing with M. tuberculosis culture | To detect resistance to first-line and/or second-line anti-TB medicines | Drug susceptibility testing | |
| | Lipoarabinomannan (LAM) antigen | Diagnosis of TB in seriously ill HIV-positive inpatients | RDT | 2015 |
| | Immune response by Interferongama Release assay (IGRA) | For diagnosis of latent TB infection | Immunoassay or ELISPOT assay | 2015 |
| Malaria | Plasmodium spp. antigens; species-specific (e.g. HRP2) and/or pan-species-specific (e.g. pan-pLDH) | For diagnosis of one or more human malaria species (P. falciparum, P. vivax, P. malariae, P. ovale) | RDT | 2006 |
| | Plasmodium spp. | For diagnosis of one or more human malaria species (P. falciparum, P. vivax, P. malariae, P. ovale) and monitoring response to treatment | Light microscopy | 2006 |
| | Glucose-6- phosphate dehydrogenase (G6PD) activity | To determine G6PDactivity (normal, intermediate, deficient) for a decision to administer 8-aminoquinoline group drugs for radical cure of P. vivax malaria | Semiquantitative fluorescent spot test | |
| Syphilis | Antibodies to Treponema pallidum | For diagnosis or as an aid in the diagnosis of syphilis | RDT | 2013 |
| | | | Immunoassay | |
| | Antibodies to T. pallidum and to HIV-1/2 (anti-HIV Ab) | For diagnosis or as an aid in diagnosis of HIV-1/2 infection and/or syphilis | RDT | 2017 |
| | Nontreponemal rapid plasma reagin (RPR) test | For screening for syphilis and monitoring treatment effectiveness | Particle/charcoal Agglutination assay | 2003 |
| | Nontreponemal venereal disease research laboratory (VDRL) test | For screening, diagnosis and confirmation of neurosyphilis | Flocculation test | |
| | T. pallidum haemagglutination (TPHA) test | For confirmation of syphilis infection & diagnosis of early & late syphilis infection | Red cell agglutination assay | 2013 |
| | T. pallidum particle agglutination (TPPA) test | | Particle agglutination assay | |
| Haematology | Complete blood count (CBC) Automated | To evaluate overall health and to detect a wide range of disorders, including anaemia, infections, leukaemias, red blood cell, white blood cell and platelet abnormalities and primary immune disorders. To diagnose and monitor chemotherapy associated myelotoxicity Note: Result time sensitive for emergency and critical care | Automated haematology analyser, total & differential counts of white blood cell (WBC), red blood cell (RBC), platelets, haemoglobin (Hb) and haematocrit (Hct) | 1989 |

Countries (Mentioned in national test menu and NEDL): Benin (2009), Botswana (2019), Burkina Faso (2009), Burundi (2016), Cameroon (2011), Cote d'Ivoire (2008), eSwatini (2010), Ethiopia (2013), Gabon (2012), Gambia (ND), Ghana (2021), Guinea (2009), Guinea Bissau (ND).

**Mentioned in national test menu and NEDL**

| Programs/ Diseases | Diagnostic test (Mentioned in WHO EDL 2) | Assay format | Test purpose | Year of WHO approval | Kenya 2014 | Liberia 2011 | Malawi 2009 | Mali 2005 | Niger ND | Nigeria 2021 | Senegal 2015 | Sierra Leone 2015 | South Africa 2018 | Tanzania 2018 | Togo 2021 | Uganda 2017 | Zimbabwe 2015 |
|---|---|---|---|---|---|---|---|---|---|---|---|---|---|---|---|---|---|
| HIV | Antibodies to HIV-1/2 (anti-HIV Ab) | RDT | For the diagnosis of HIV infection: adults, adolescents, children and infants > 18 months of age | 1999 | | | | | | | | | | | | | |
| | | Immunoassay | | | | | | | | | | | | | | | |
| | Combined HIV antibody/p24 antigen (anti-HIV/p24 Ag) | RDT | For the diagnosis of HIV infection: adults, adolescents, children and infants > 18 months of age | | | | | | | | | | | | | | |
| | | Immunoassay | | | | | | | | | | | | | | | |
| | Qualitative HIV virological nucleic acid test | Nucleic acid test | For diagnosis of HIV infection in infants < 18 months of age | | | | | | | | | | | | | | |
| | Quantitative HIV virological nucleic acid test | Nucleic acid test (DBS) | For monitoring response to antiviral treatment. For diagnosis of HIV infection in infants < 18 months of age (only if validated by the manufacturer) | | | | | | | | | | | | | | |
| | CD4 cell enumeration | Flow cytometry | For staging advanced HIV disease For monitoring response to antiretroviral therapy. (In settings where viral load is not available) | 2009 | | | | | | | | | | | | | |
| | Cryptococcal antigen | RDT | For screening and diagnosis of cryptococcal meningitis in people with advanced HIV disease | 2009 | | | | | | | | | | | | | |
| | | Immunoassay | | 2009 | | | | | | | | | | | | | |
| | Histoplasma antigen | Immunoassay | To aid in the diagnosis of disseminated Histoplasmosis | 2017 | | | | | | | | | | | | | |
| TB | Mycobacterium tuberculosis bacteria | Microscopy | For diagnosis, treatment and monitoring of active TB | 2008 | | | | | | | | | | | | | |
| | | Bacterial culture | For diagnosis and treatment monitoring of active TB including drug resistant TB | | | | | | | | | | | | | | |
| | M. tuberculosis DNA | Nucleic acid test | For diagnosis of active TB and simultaneous detection of rifampicin resistance | 2008 | | | | | | | | | | | | | |
| | | Loop-mediated isothermal amplification (LAMP) | For diagnosis of active TB | | | | | | | | | | | | | | |
| | M. tuberculosis DNA mutations associated with resistance | Molecular line probe assay (LPA) | For detection of resistance to first-line anti-TB medicines | | | | | | | | | | | | | | |
| | | Molecular line probe assay (LPA) | For detection of resistance for second-line anti-TB medicines | | | | | | | | | | | | | | |
| | Drug susceptibility testing with M. tuberculosis culture | Drug susceptibility testing | To detect resistance to first-line and/or second-line anti-TB medicines | | | | | | | | | | | | | | |
| | Lipoarabinomannan (LAM) antigen | RDT | Diagnosis of TB in seriously ill HIV-positive inpatients | 2015 | | | | | | | | | | | | | |
| | Immune response by Interferongma Release assay (IGRA) | Immunoassay or ELISPOT assay | For diagnosis of latent TB infection | 2015 | | | | | | | | | | | | | |
| Malaria | Plasmodium spp. antigens; species-specific (e.g. HRP2) and/or pan-species-specific (e.g. pan-pLDH) | RDT | For diagnosis of one or more human malaria species (P. falciparum, P. vivax, P. malariae, P. ovale) | 2006 | | | | | | | | | | | | | |
| | Plasmodium spp. | Light microscopy | For diagnosis of one or more human malaria species (P. falciparum, P. vivax, P. malariae, P. ovale) and monitoring response to treatment | 2006 | | | | | | | | | | | | | |
| | Glucose-6-phosphate dehydrogenase (G6PD) activity | Semiquantitative fluorescent spot test | To determine G6PD activity (normal, intermediate, deficient) for a decision to administer 8-aminoquinoline group drugs for radical cure of P. vivax malaria | | | | | | | | | | | | | | |
| Syphilis | Antibodies to Treponema pallidum | RDT | For diagnosis or as an aid in the diagnosis of syphilis | 2013 | | | | | | | | | | | | | |
| | | Immunoassay | | | | | | | | | | | | | | | |
| | Antibodies to T. pallidum and to HIV-1/2 (anti- HIV Ab) | RDT | For diagnosis or as an aid in diagnosis of HIV-1/2 infection and/or syphilis | 2017 | | | | | | | | | | | | | |
| | Nontreponemal rapid plasma reagin (RPR) test | Particle/charcoal Agglutination assay | For screening for syphilis and monitoring treatment effectiveness | 2003 | | | | | | | | | | | | | |
| | Nontreponemal venereal disease research laboratory (VDRL) test | Flocculation test | For screening, diagnosis and confirmation of neurosyphilis | | | | | | | | | | | | | | |
| | T. pallidum haemagglutination (TPHA) test | Red cell agglutination assay | For confirmation of syphilis infection & diagnosis of early & late syphilis infection | 2013 | | | | | | | | | | | | | |
| | T. pallidum particle agglutination (TPPA) test | Particle agglutination assay | | | | | | | | | | | | | | | |
| Haematology | Complete blood count (CBC) Automated | Automated haematology analyser, total & differential counts of white blood cell (WBC), red blood cell (RBC), platelets, haemoglobin (Hb) and haematocrit (Hct) | To evaluate overall health and to detect a wide range of disorders, including anaemia, infections, leukaemias, red blood cell, white blood cell and platelet abnormalities and primary immune disorders. To diagnose and monitor chemotherapy associated myelotoxicity Note: Result time sensitive for emergency and critical care | 1989 | | | | | | | | | | | | | |

Green: Test and test assay format mentioned in document

Rose: only diagnostic test or test purpose mentioned in document

Grey: No mention of the test in document

The main reported challenge in developing an NEDL or revision of test menus is funding. Respondents noted they need funding for the national laboratory landscape survey, consultants, stakeholder workshops and dissemination meetings.

Respondents already envisioned what could contribute to possible success or failure to implementation of an NEDL. They perceive that successful implementation of NEDL is closely related to how the NEDL has been developed. If the development has considered the laboratory system bottlenecks, identified by mapping the laboratory landscape, and has involved all relevant stakeholders in development, the implementation will be easier. Involvement of laboratory professionals and medical stakeholders is important so that they feel ownership of the NEDL and commitment to implement: "*. . .. because none of them* [will] *see the document for the first time*, [because they] *have been really involved in the development*" as Nigerian respondent noted. They consider that regulatory agencies are key-stakeholders since they will ensure that laboratories are staffed only by qualified personnel. For successful implementation of the NEDL, it is important to improve the quality management system that includes equipment, human resources training, and structure, and ensure adequate procurement of IVDs listed on the NEDL.

## Discussion

The study documents whether and how countries currently organize the delivery of essential diagnostics across their tiered laboratory network, with highlights on challenges and opportunities for ensuring the availability of and access to laboratory tests for effective healthcare. While NEDL has the potential to guide rational investment in diagnostics, our data suggest that the effective development and implementation of NEDL will require various adjustments and ancillary system interventions including (i) strong linkages between NEDL and existing laboratory strategies and operational plans; (ii) the institutionalization of community testing activities within the structure of national laboratory network; and (iii) the increased involvement of the laboratory sector in health policy decision making.

### NEDL to add value to existing diagnostic strategies

Many countries already have national test menus. However, these are often outdated and, mainly for budgetary reasons, have not been revised. These test menus specify IVDS by health care/laboratory tier and many are more practical in terms of operationalization than a standard NEDL, because many also include equipment, consumables, and sometimes laboratory personnel and infrastructure requirements. An NEDL would de facto be part of such a test menu.

WHO recommends that countries adapting the WHO EDL should add granularity and specify tests by level of the health-care system appropriate to the local context [12]. The Nigerian NEDL did not follow this recommendation, but adapted the two-tiers WHO EDL, thereby posing the question of the likelihood of operationalization of this strategy. Making laboratory tier-specific test guidelines (as the national test menus do) is more practical and realistic considering the context of weak laboratory services in many LIMC, especially at peripheral level–as qualitative study findings corroborate. For example: The Nigerian NEDL enlists a total of 145 IVDs for clinical settings covering primary, secondary, tertiary and national reference laboratories and 7 IVDs for screening of blood donations. Given the generally insufficient numbers and unqualified human resources [1] usually observed in LMIC, lumping tests together in the NEDL regardless the tier level represent a missed opportunity to guide laboratory staff in charge of peripheral laboratories (many lacking proper training) to select tests appropriate for a specific tier level. National test menu, on the other hand, usually contain more specific and prescriptive information on what test to use and where and where not. To be 'implementable'

an NEDL should be accompanied by operational documents, either in separate sections of an NEDL or in complementary guidelines, as recommended by WHO. Some of the test menus include part of these operational guidelines.

Respondents' motivation to develop an NEDL or revise their test menu stemmed from the need to solve or alleviate some of the present problems in the laboratory system, including over- and under-stocking and -supply of certain IVDs, erratic procurement and supply by multiple vendors, inefficient resource allocation, and problems in regulating private laboratories. They believed an NEDL or revision of their test menu would oil the IVD supply chain by guiding all those involved, including national regulatory bodies, planners, public and private procurers, funders, and public and private health facilities. Countries also saw the importance of international/regional harmonisation of essential IVDs, because this could facilitate them lobbying for resources at international level. In addition, countries having the same IVDs on their lists could motivate setting up manufacturing at a regional (African) level.

## Community testing as an integral part of national testing strategies

The WHO EDL specifies tests for the "community tier" to increase access to testing. This provides the opportunity for non-laboratory staff to conduct rapid diagnostic tests during outreach in the community and in health posts or health centres without a laboratory on-site. The Nigerian NEDL lists 12 general IVDs and 15 disease specific IVDs (diseases include Cholera; Hepatitis B and C; HIV; Malaria; Syphilis; Tuberculosis; Peptic Ulcer) for this "community tier", in contrast to most of the current test menus, which do not specify tests for this community tier (n = 5 only).

Quality requirements such as compliance to the ISO 15189 are reportedly not universally enforced in all clinical laboratory [21], let alone in testing sites outside conventional laboratory facility. For this reason testing activities taking place at community level, "point of care" or "point of need" often raise concerns about their compliance to quality assurance, or biosafety/biosecurity requirements such as waste management among other issues. In the recently published ISO 15189:2022 the requirements and regulations for point-of-care testing (previously in ISO 22870) have been incorporated, opening the route for more quality and safety of community level testing [20], as part of national laboratory quality strategies [21, 22].

Testing at community level is often termed "task-shifting"; simple tests (such as RDT) are conducted by non-laboratory personnel (lay workers) in the community and health post/centres without laboratory on-site. In fact many vertical disease programmes use this system. The Zimbabwe respondents explained: ''*HIV and malaria tests are task-shifted to the community and rural or urban health clinics without a laboratory* [and laboratory personnel] *on site. Tests are done by nurses, primary counsellors, and community health workers. Sometimes they also do rapid testing for syphilis, and nowadays COVID-19. The health centres with a laboratory distribute the tests and consumables to these workers. In Zimbabwe exists an integrated sample transportation system, where samples that are taken at lower levels move to higher levels, and patients do not have to move. This increases access.*'' In reality, many countries have yet not integrated community testing as an outreach function of the laboratory network (except for vertical programmes) with insufficient resources, roles and responsibilities for supervision of the lay workers. A respondent in Nigeria explaining: *"Labs are defending their territories"*, providing some clues as why this much needed integration is not happening.

A clear governance, coordination and regulation of community testing by the laboratory authorities is warranted. Rather than task shifting, we propose the term "task sharing", for testing at community level, coordinated and supervised by laboratory trained personnel, also carrying a less "threatening" connotation for the laboratory sector.

## Aligning NEDL to the local context of the national laboratory system

In order to increase the likelihood of implementation, WHO advises that NEDL should factor in the pyramidal structure and overall requirements of national health care and laboratory system and align to larger national health policies [12]. However, respondents warn that the respective level of clinical and laboratory facilities do not always match within individual hospitals. In addition, given that regulatory bodies assign tier level for laboratory tiers mainly for licencing purposes (and because licensure of public facilities cannot be withdrawn), the theoretical and actual capacity of laboratories are often far apart.

Conducting a laboratory landscape survey, including study of existing IVDs, laboratory personnel, equipment, consumables and barriers to access like was done by Nigeria is a useful exercise, allowing the country to tailor the NEDL to constraints and opportunities of their health and laboratory system. In this way the respondents considered their NEDL a more realistic and implementable list and not an aspirational list. Depending on the complexity of the national laboratory system requirement, countries should consider developing additional operational describing how to comply minimal requirements for equipment, commodities or staff's competencies and number at each tier; or best alternative when requirements are not met.

In addition to the laboratory landscape, a review of national epidemiological bulletins, disease program data, surveillance report and other available health data might provide an opportunity to refine the regional and national estimates of disease burden and priority pathogen, which are critical to the rational selection of IVD.

## Stakeholder's inclusiveness and laboratory governance decision making power for NEDL development and roll out

Our data suggest the positive role of implementing partners, clinicians and laboratory staff at different tiers of the health care pyramid, and regulatory and litigation agencies in working groups and workshops, in the development of comprehensive essential diagnostic test strategies. The Nigerian experience showed that when all are involved in decision-making discussions, they feel committed and take ownership and may already start implementing before the NEDL is officially launched. The Nigerian Minister of Health had played a key role in getting off the development and launch of the NEDL.

The Minister of Health and/or State Secretary of Health should be the foremost stakeholders to commit themselves to the development of an NEDL. In the situation whereby National Laboratory Directorate (when they exist) have little decision-making power and insufficient budget [4, 5], the Minister of Health and/or State Secretary of Health represent a stronger political force, which can appoint NEDL committee also beyond the remit of laboratory medicine (such as clinicians, funders, or civil society) and can source for domestic or external funding more easily. The Minister of Health might have the adequate political traction to identify, advocate for and mobilize funding within vertical disease programmes, international aid, insurance companies, out-of-pocket health budget and private sectors, which can all contribute to the actual implementation of the NEDL necessary for the long term.

The Minister of Health should designate the Directorate of Laboratory Services or similar governance unit to lead the NEDL development process, implementation and continuous improvement process on their behalf. Using geographic information system data on laboratory testing capacity such as the ASLM LabMap [23], provide a unique opportunity to continuously monitor the availability of IVDs at each tier of the national laboratory network

## Recommendations

Recommendations by study participants and the authors address countries and partners such as ASLM, FIND, and WHO intending to support NEDL development and implementation (Table 8). It should be noted that these recommendations are partly in WHO guidelines.

**Table 8. Recommendations for NEDL development and implementation.**

| Proposed areas for improvement | Recommendations | Stakeholders concerned |
|---|---|---|
| **NEDL to add value to existing diagnostic strategies and contain sufficient information for operationalization** | 1. Countries should make an informed decision on whether they want to develop an NEDL or take it further and go for revision and improvement of their test menu, also given the context of limited resources for policy development<br>2. NEDL or test menu should go beyond the generic WHO EDL guidance only referring to "facility with/without laboratory" and clearly assign a laboratory tier level to all IVDs | 1. MoH<br>2. NEDL committee, WHO |
| **Community testing is part of national testing strategies and "community tier" is an extension of national laboratory network.** | 3. Improve the utility of national test menu by incorporating a section for the "community tier"<br>4. Define human resources, quality assurance, biosafety requirement and supervision mechanisms by higher tier, for all testing services conducted at community level. Refer to relevant sections of ISO 15189:2022 [20].<br>5. Establish clear governance, coordination and regulation of community testing services, re branded as "task-sharing"<br>6. Reviewing how vertical programmes include community testing in their diagnostic services can provide useful examples to expand apply at national level. | 3. Directorate of Laboratories, MoH<br>4. NEDL committee, Directorate of Laboratory, National Standard Agency, Laboratory Quality Agencies<br>5. MoH, Directorate of Laboratories<br>6. HIV, TB and Malaria programmes, Directorate of Laboratories |
| **Aligning NEDL to the local context and situation of the national laboratory system context** | 7. Conduct a laboratory system landscape analysis to inform the development of the NEDL.<br>8. Review local health data to refine estimates of disease burden and priority pathogen whenever possible<br>9. Criteria for IVD selection should consider the requirements and realities of the health care and laboratory system, in support of a realistic rather than an aspirational NEDL. Important considerations include:<br> i. The national policy legal and regulatory framework<br> ii. The constraints of the supply chain and the national procurement systems<br> iii. The health insurance systems and coverage of diagnostic costs<br> iv. The existence and performance of a specimen referral system<br> v. Current barriers to access of health care services<br>10. Specify priority IVDs by laboratory tier (including the "community tier") and not by health system tier.<br>11. Develop additional operational guidelines to facilitate the implementation of the NEDL /revised test menu at each tier of the laboratory system and in the context of compliance challenges. | 7. MoH, implementing partners, funders<br>8. MoH, NEDL committee, Directorate of Laboratories, WHO, researchers, public health specialists<br>9. MoH, NEDL committee<br>10. 11. Directorate of Laboratories, MoH, national laboratory technical working group |
| **Stakeholder's inclusiveness and increased decision making power of the laboratory governance for NEDL development and roll out** | 12. Include a large representation of laboratory, clinical, technical and implementing partners and civil society stakeholders in the development of the NEDL or revision of test menu.<br>13. Conduct the process under the leadership of the Minister of Health, subsequently appointing the National Laboratory Directorate to coordinate the development and roll out of the NEDL<br>14. Include various sources of funding (domestic, private, programmatic, out of pocket, health insurance premiums) into the costing and financial plan for the roll out of NEDL<br>15. Plan for the collection of relevant national data to assess the progress of the NEDL implementation and take corrective actions (e.g. ASLM and Africa CDC LabMap data) | 12. MoH<br>13. MoH, Directorate of Laboratories<br>14. MoH<br>15. MoH, Directorate of Laboratories, national health statistics, ASLM, Africa CDC |

### Study limitations

In recommendations for developing and implementing an NEDL, countries could learn from the general adoption and use of the WHO essential medicine list; it was beyond the present study to analyse the literature on this topic.

## Conclusion

The findings of this study provide a unique set of evidence-based practical recommendations to countries and partners supporting the laboratory system on the processes of development and effective implementation of NEDL in Africa. The information analysed here provides a unique perspective that can inform the development and implementation of an NEDL or similar guidelines for tier-specific essential IVDs.

## Supporting information

**S1 Table. Test menus.**
(PDF)

**S2 Table. Country documents.**
(PDF)

**S3 Table. Additional country documents.**
(PDF)

**S4 Table. Data for spearman correlation test.**
(PDF)

**S1 Data. Quantitative data set.**
(ZIP)

**S2 Data. Summary of qualitative data by country.**
(PDF)

**S1 Fig. Flowchart sampling of documents.**
(PDF)

## Acknowledgments

We are grateful to the 43 respondents from Burkina Faso, Cameroon, Ethiopia, Kenya, Nigeria, Uganda and Zimbabwe for agreeing to participate in individual or group interviews despite their busy schedules. They shared their knowledge, views of and rich experiences with their national laboratory systems, essential diagnostics and test menus. We thank the LabCop country contact persons who shared their national test menus that we did not find online. We also thank people from the ASLM and FIND teams that contributed their insights.

## Author Contributions

**Conceptualization:** Francis Ocen, Kekeletso Kao, Pascale Ondoa.

**Data curation:** Elishebah Maruta Mutegi, Francis Ocen, Maina Michael Waweru, Sri Lakshmi Priyanka Yerra, Jenny L. Grunwald, Lucy A. Perrone.

**Formal analysis:** Winny Koster, Elishebah Maruta Mutegi, Francis Ocen, Maina Michael Waweru, Albert Gautier Ndione, Sri Lakshmi Priyanka Yerra, Jenny L. Grunwald, Lucy A. Perrone, Pascale Ondoa.

**Funding acquisition:** Kekeletso Kao.

**Investigation:** Winny Koster, Elishebah Maruta Mutegi, Francis Ocen, Collins Otieno Odhiambo, Albert Gautier Ndione, Sri Lakshmi Priyanka Yerra, Jenny L. Grunwald, Delores Mack, Lucy A. Perrone.

**Methodology:** Winny Koster, Elishebah Maruta Mutegi, Francis Ocen, Albert Gautier Ndione, Kekeletso Kao, Pascale Ondoa.

**Project administration:** Francis Ocen, Collins Otieno Odhiambo, Maina Michael Waweru, Pascale Ondoa.

**Resources:** Elishebah Maruta Mutegi, Sri Lakshmi Priyanka Yerra, Jenny L. Grunwald, Lucy A. Perrone.

**Supervision:** Collins Otieno Odhiambo, Kekeletso Kao, Lucy A. Perrone, Pascale Ondoa.

**Validation:** Winny Koster, Collins Otieno Odhiambo, Lucy A. Perrone.

**Visualization:** Winny Koster, Elishebah Maruta Mutegi, Albert Gautier Ndione, Jenny L. Grunwald, Lucy A. Perrone, Pascale Ondoa.

**Writing – original draft:** Winny Koster, Pascale Ondoa.

**Writing – review & editing:** Winny Koster, Elishebah Maruta Mutegi, Francis Ocen, Collins Otieno Odhiambo, Maina Michael Waweru, Albert Gautier Ndione, Sri Lakshmi Priyanka Yerra, Jenny L. Grunwald, Delores Mack, Kekeletso Kao, Lucy A. Perrone, Pascale Ondoa.

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
