## [Decision Letter · Decision Letter 0]

17 Mar 2023

PGPH-D-22-02094

Contexts for developing of national essential diagnostics lists

Lessons from a mixed-methods study of existing documents, stakeholders and decision making on tier-specific essential in-vitro diagnostics in African countries

Dear Dr. Koster,

Thank you for submitting your manuscript to PLOS Global Public Health. After careful consideration, we feel that it has merit but does not fully meet PLOS Global Public Health’s publication criteria as it currently stands. Therefore, we invite you to submit a revised version of the manuscript that addresses the points raised during the review process.

We look forward to receiving your revised manuscript.

Kind regards,

Shifa S. Habib

Academic Editor

Journal Requirements:

2. Please send a completed 'Competing Interests' statement, including any COIs declared by your co-authors. If you have no competing interests to declare, please state "The authors have declared that no competing interests exist". Otherwise please declare all competing interests beginning with the statement "I have read the journal's policy and the authors of this manuscript have the following competing interests:"

3. Please amend your detailed Financial Disclosure statement. This is published with the article. It must therefore be completed in full sentences and contain the exact wording you wish to be published.

4. In the online submission form, you indicated that "The quantitative database and analysis can be accessed by requesting the authors working for ASLM. The qualitative data transcripts and analysis can be accessed through the first (corresponding) author". All PLOS journals now require all data underlying the findings described in their manuscript to be freely available to other researchers, either 1. In a public repository, 2. Within the manuscript itself, or 3. Uploaded as supplementary information.

5. We noticed that you used "data not shown" in the manuscript. We do not allow these references, as the PLOS data access policy requires that all data be either published with the manuscript or made available in a publicly accessible database. Please amend the supplementary material to include the referenced data or remove the references.

Additional Editor Comments (if provided):

Reviewers' comments:

Reviewer's Responses to Questions

**Comments to the Author**

1. Does this manuscript meet PLOS Global Public Health’s publication criteria? Is the manuscript technically sound, and do the data support the conclusions? The manuscript must describe methodologically and ethically rigorous research with conclusions that are appropriately drawn based on the data presented.

Reviewer #1: Yes

Reviewer #2: Yes

2. Has the statistical analysis been performed appropriately and rigorously?

Reviewer #1: N/A

Reviewer #2: I don't know

3. Have the authors made all data underlying the findings in their manuscript fully available (please refer to the Data Availability Statement at the start of the manuscript PDF file)?

Reviewer #1: No

Reviewer #2: No

4. Is the manuscript presented in an intelligible fashion and written in standard English?

Reviewer #1: Yes

Reviewer #2: Yes

5. Review Comments to the Author

Reviewer #1: Koster et al conducted a mixed-methods study aimed at collecting information from existing national guidance documents from different African countries and on decision making when selecting tier-specific IVDs that could be the basis for recommendations to supporting countries in developing an NEDL or a similar guidance document. They examined xxx documents from 48 countries and conducted key informant interviews with 43 participants from 7 countries. Based on this they provide recommendations concerning development of the NEDL, including a community tier in the NEDL, embedding the NEDL into the national laboratory system context, and stakeholder involvement. These recommendations will be useful as more countries are expected to adapt the WHO EDL to suit national needs.

Specific comments

Introduction

Line 65: ….unique set of

Line 69: Replace ‘treatment’ with ‘execution’ or ‘implementation’ or other suitable word.

Line 81: Remove ‘A’.

Line 82: I believe the word should be ‘means’ and not ‘makes’

Line 88: Remove the parenthesis from the word ‘sufficiently’

Line 111: The abbreviation UHC was not stated in Line 68 where Universal Health Coverage was first used

Materials and methods

Lines 195-204: Line 50: It appears the number of documents reviewed were 293, however in line 50, from the abstract, it is stated that ‘362 documents from 48 African countries were reviewed….’. A flow chart showing the document elimination and selection process and a breakdown of document types would be useful.

A supplementary reference list of the URLs for those documents which could be accessed online should be provided to satisfy the journal’s data availability requirements.

Discussion

Authors should create a Figure Box or table with the recommendations enabling them to be seen at a glance.

Reviewer #2: I must express my sentiments of gratitude to the editor for inviting me for this peer-review, and also appreciate the authors for putting in so much effort in this work. The problem statement was well-defined and highlights the current gaps in existing essential diagnostic list, which not in tune with the current disease burden with respect to the African countries studied, as well as uncovering the limitations in terms of compliance with regulatory guidelines and the various factors contributing to this. The authors showed an in-depth literature review of the current state of NEDL in Africa. Study objectives were also clearly stated WITHOUT ambiguity.

Are there specific relatable instances (with statistics) where the outcome of a healthcare management program was impacted by the lack of a robust NEDL or test menus? For example, in the management of outbreaks such as cholera, Ebola, COVID-19, etc. This will further drive home the need to address the problem statement.

In the Materials and Methods section, particularly in the Sampling and sample size of countries section (from lines 191 – 206); Will it be possible to present the distribution by category of all the different documents sourced using a simple table or a chart, so readers can get a snapshot of the information, in addition to the written text?

The result of the study clearly uncovered the strengths and weaknesses of the different current NEDL and test menu documents in the countries studied, as well as the factors responsible for the gaps in development and implementation. All recommendations made in the discussion were based on the results: such as embedding the NEDL in the national laboratory system context, especially because the test menus (which were country specific) were found to be easily operationalized and in tune with the strengths and weaknesses of country’s health system, the inclusion of relevant stakeholders with decision making power and influence in terms of operationalization and compliance in the formulation of the policy documents.

However, in addition to the recommendations, will it be possible to include example of how other LMICs from other continents (e.g Asia, South America, etc) have managed their NEDL and their current status as far as effectiveness. This can serve as a benchmark for African governments and policy makers to measure how well their countries have performed in terms of implementation of IVDs or how much work needs to done in order to address these deficiencies.

6. PLOS authors have the option to publish the peer review history of their article (what does this mean?). If published, this will include your full peer review and any attached files.

**Do you want your identity to be public for this peer review?** For information about this choice, including consent withdrawal, please see our Privacy Policy.

Reviewer #1: **Yes: **Iriagbonse Iyabo Osaigbovo

Reviewer #2: No

---

## [Editor Report · Decision Letter 1]

27 Apr 2023

Contexts for developing of national essential diagnostics lists

Lessons from a mixed-methods study of existing documents, stakeholders and decision making on tier-specific essential in-vitro diagnostics in African countries

PGPH-D-22-02094R1

Dear Dr. Koster,

We are pleased to inform you that your manuscript 'Contexts for developing of national essential diagnostics lists

Lessons from a mixed-methods study of existing documents, stakeholders and decision making on tier-specific essential in-vitro diagnostics in African countries' has been provisionally accepted for publication in PLOS Global Public Health.

Best regards,

Shifa S. Habib

Academic Editor